# New Insight into the Genotype-Phenotype Correlation of *PRPH2*-Related Diseases Based on a Large Chinese Cohort and Literature Review

**DOI:** 10.3390/ijms24076728

**Published:** 2023-04-04

**Authors:** Yingwei Wang, Junwen Wang, Yi Jiang, Di Zhu, Jiamin Ouyang, Zhen Yi, Shiqiang Li, Xiaoyun Jia, Xueshan Xiao, Wenmin Sun, Panfeng Wang, Qingjiong Zhang

**Affiliations:** State Key Laboratory of Ophthalmology, Zhongshan Ophthalmic Center, Sun Yat-sen University, Guangdong Provincial Key Laboratory of Ophthalmology and Visual Science, Guangzhou 510060, China

**Keywords:** *PRPH2*, retinitis pigmentosa, macular degeneration, genotype-phenotype correlation

## Abstract

Variants in *PRPH2* are a common cause of inherited retinal dystrophies with high genetic and phenotypic heterogeneity. In this study, variants in *PRPH2* were selected from in-house exome sequencing data, and all reported *PRPH2* variants were evaluated with the assistance of online prediction tools and the comparative validation of large datasets. All variants were classified based on the American College of Medical Genetics and Genomics/Association for Molecular Pathology (ACMG/AMP) guidelines. Individuals with pathogenic or likely pathogenic variants of *PRPH2* were confirmed by Sanger sequencing. Clinical characteristics were summarized. Ten pathogenic or likely pathogenic variants of *PRPH2* were identified in 14 families. In our cohort, the most frequent variant was p.G305Afs*19, accounting for 33.3% (5/15) of alleles, in contrast to the literature, where p.R172G (11.6%, 119/1028) was the most common variant. Nine in-house families (63.8%) were diagnosed with retinitis pigmentosa (RP), distinct from the phenotypic spectrum in the literature, which shows that RP accounts for 27.9% (283/1013) and macular degeneration is more common (45.2%, 458/1013). Patients carrying missense variants predicted as damaging by all seven prediction tools and absent in the gnomAD database were more likely to develop RP compared to those carrying missense variants predicted as damaging with fewer tools or with more than one allele number in the gnomAD database (*p* = 0.001). The population-specific genetic and phenotypic spectra of *PRPH2* were explored, and novel insight into the genotype–phenotype correlation of *PRPH2* was proposed. These findings demonstrated the importance of assessing *PRPH2* variants in distinct populations and the value of providing practical suggestions for the genetic interpretation of *PRPH2* variants.

## 1. Introduction

Peripherin 2 (*PRPH2*), mapped at 6p21.2 and initially named Retinal Degeneration Slow (*RDS*), encodes a transmembrane protein located at the disk membranes of both rod and cone photoreceptors and has an important role in morphogenesis [1,2]. Variants of *PRPH2* have been reported to be a common cause of inherited retinal dystrophies [3,4,5]. Multiple clinical manifestations contribute to the phenotypic heterogeneity of *PRPH2*-associated retinopathy, including pattern dystrophy (PD; OMIM 169150) [6]; vitelliform macular dystrophy (VMD; OMIM 608161) [7]; retinitis pigmentosa (RP; OMIM 608133) [8]; cone-rod dystrophy (CRD); and central areolar choroidal dystrophy (CACD; OMIM 613105) [9]. *PRPH2*-associated retinopathies are most often inherited as an autosomal dominant trait. In rare cases, it is inherited in the autosomal recessive mode or as a digenic form in conjunction with a variant of *ROM1*, resulting in more severe phenotypes such as Leber congenital amaurosis (LCA; OMIM 608133) and early-onset RP [10,11]. The genetic and phenotypic heterogeneity of *PRPH2*-related diseases presents certain diagnostic challenges, and the genotype–phenotype correlation of *PRPH2* is still unclear despite the large amount of research has been conducted to explore this correlation.

Clinical spectrum and genotype–phenotype correlation analyses of *PRPH2* have been mostly carried out in Caucasian cohorts; *PRPH2*-associated retinopathy has been rarely studied in Asian cohorts, particularly in Chinese cohorts. The contribution of *PRPH2* has been reported to be 4.6% for inherited retinal dystrophies in European descendants and 3.6% for RP patients in Spain. However, in a previous cohort of Chinese patients with suspected RP, the contribution of *PRPH2* was found to be as low as 0.06%, raising a potential ethnic difference compared to the relatively high variant frequency in European descendants [3,12,13,14]. In this study, variants of *PRPH2* were collected from our large exome sequencing data and thoroughly analyzed by performing multistep bioinformatics, comparing the data with a large general population gnomAD database, and classifying the results according to the ACMG/AMP guidelines. The clinical characteristics of individuals with pathogenic or likely pathogenic variants of *PRPH2* were summarized. In total, eight families with autosomal dominant RP, one family with autosomal recessive RP, and five families with MD (including two in a digenic pattern) were identified in our cohort. Among our 1519 in-house RP families, the contribution of *PRPH2* was approximately 0.6% (9/1519), potentially making it one of the top six implicated genes for autosomal dominant RP in the Chinese cohort. Additionally, reported variants of *PRPH2* (including those recorded in the Human Gene Mutation Database (HGMD) database and in recent literature) and related phenotypes have been summarized. All variants were assessed in detail using the same procedures as our in-house variants, and available clinical data from patients were compared with that of our in-house patients. It was found that the main phenotype of *PRPH2* in the Asian population was RP, while the main phenotype in Caucasians is more likely to be MD, which demonstrated an ethnic-specific genotypic and phenotypic spectrum of *PRPH2* between Caucasian populations and Asian subjects. Based on a comprehensive analysis of variants by the ACMG/AMP guidelines, it was found that patients with the missense variants predicted as damaging with more in silico tools may have more severe phenotypes and an earlier age of onset. The genotype–phenotype correlation was explored, and suggestions for the evaluation of pathogenic variants of *PRPH2* are proposed, which can help enhance our understanding of *PRPH2*.

## 2. Results

### 2.1. In-House PRPH2 Variants Identification

A total of 49 variants of *PRPH2* were detected, including 44 missense variants, three frameshift variants (c.914del/p.G305Afs*19, c.595_607del/p.N199Gfs*53, and c.264dup/p.A89Sfs*88), one nonsense variant (c.423C>G/p.Y141*), and one in-frame variant (c.499_504dup/p.G167_N168dup). In addition to the four truncation variants and one in-frame variant, 12/44 missense variants were studied and are considered potential pathogenic variants; they were predicted to be damaging by no less than five out of the seven in silico tools and had one (c.584G>T) or no allele count recorded in the gnomAD dataset (Table 1). All 17 variants in *PRPH2* were confirmed by Sanger sequencing followed by a co-segregation analysis among the available family members (Appendix A).

### 2.2. Classification of In-House PRPH2 Variants

After assessment according to the ACMG/AMP guideline, ten variants in *PRPH2*, including two truncation variants and eight missense variants, were classified as pathogenic or likely pathogenic and identified in 14 families with *PRPH2*-associated retinopathy, including 11 newly recruited families and three reported families in our previous studies [15] (Figure 1). Of them, 9/14 were diagnosed with RP, eight of which had RP with classic autosomal dominant inheritance, caused by the truncation of p.G305Afs*19 and three missense variants (c.797G>A/p.G266D, c.633C>G/p.F211L, and c.535T>C/p.W179R, respectively); additionally, one patient had autosomal recessive RP caused by the variant c.452T>G/p.F151C in homozygous status. In total, 5/14 families were diagnosed with MD, including one proband who carried the variant p.Y141* and presented with late-onset Stargardt disease (STGD). Two patients were diagnosed with adult vitelliform macular dystrophy (AVMD) caused by one *PRPH2* variant in conjunction with one potential pathogenic variant in *ROM1*. Moreover, two unrelated patients carried the variants c.584G>T/p.R195L and c.584G>A/p.R195Q, respectively, and showed MD. The remaining seven variants were considered variants of uncertain significance and were detected in seven families that showed phenotypes unrelated to *PRPH2*, including 2/7 probands no older than 10 years old who carried truncation variants and were diagnosed with congenital ectopia lentis and retinoblastoma, respectively; 3/7 with missense variants presented with early-onset high myopia and macular coloboma; 1/7 probands carrying the in-frame variant was diagnosed with primary open angle glaucoma; and 1/7 individuals did not show any ocular abnormalities (Appendix A). Among the ten pathogenic or likely pathogenic variants, p.G305Afs*19 was the most frequently observed variant with an allele frequency of 33.3% (5/15).

### 2.3. Phenotype Characterization of In-House Patients

Of the 12 patients from nine families with *PRPH2*-associated RP, including eight families with the autosomal dominant trait and one family with the autosomal recessive trait, 6/12 reported experiencing decreased vision or visual field defects in the third to sixth decade of life, and 4/12 reported experiencing night blindness or poor vision since childhood. Ten of them had available best-corrected visual acuity (BCVA) results, in which 9/10 patients had BCVA no less than 0.3 and only 1/10 had BCVA of hand movement. The fundi of ten patients presented with typical RP, i.e., a pale optic disc, narrow retinal arterioles, widespread tapetoretinal degeneration, macular atrophy, and midperipheral bone-spicule pigment deposits (Figure 2A–D). Full-field electroretinogram (ERG) was performed in five patients and showed undetectable or severely reduced responses of both rods and cones. Optical coherence tomography (OCT) examinations performed in three patients presented changes suggestive of photoreceptor degeneration predominantly in the perifoveal area. The constriction of the visual field was recorded in two probands.

Of the five probands with *PRPH2*-associated MD, including three in an autosomal dominant form and two with a digenic pattern, all suffered from vision problems in the fourth to fifth decades and had a BCVA of no more than 0.32. In the fundus examination, one female patient with STGD showed multiple yellowish irregular flecks scattered around the posterior pole with macular atrophy, one female patient with AVMD showed a small round atrophic lesion in the macula with a hypoautofluorescent lesion, as observed in the fundus autofluorescence image (Figure 2E), and three female patients presented with retinal pigment epithelium mottling diffusely throughout the macular area that showed widespread mottled hyper-fluorescein in the late stage of fluorescein fundus angiography (Figure 2F). Among three individuals with ERG results, 1/3 with STGD had normal cone and rod responses, and 2/3 had moderately reduced cone responses and normal rod responses. The OCT performance of five probands showed a thinning of the nuclear layer and discontinuity of the ellipsoid zone of the macular area. Central visual field defects were found in one patient. The clinical results of 17 patients from 14 families are summarized in Appendix A.

### 2.4. Genetic and Phenotypic Spectrum of PRPH2 in Different Populations

In the literature, 284 variants in *PRPH2* have been reported, including 107 truncation variants (including 32 nonsense, 61 frameshifts, 12 canonical splicing site changes, and 2 start loss variants), 149 missense variants, 21 in-frame variants, six gross deletion or insertion variants, and one 5′ untranslated variant. The most frequent variant of *PRPH2* based on the published data is c.514C>T/p.R172G with an allele frequency of 11.6% (119/1028), which is different from the most common variant p.G305Afs*19 in our cohort. The variant c.828+3A>T was the most frequent truncation variant with an allele frequency of 6.6% (68/1028) (Figure 3A, Appendix A). The truncation variants in *PRPH2* were distributed across the entire coding frame, while 119/149 (79.9%) missense variants were clustered in the intradiscal D2 loop (ID2) of the peripherin protein. As was the case for our in-house detected missense variants, the 149 missense variants were evaluated with the assistance of in silico tools and comparative validation with the gnomAD database, and finally assessed again according to the ACMG/AMP guidelines. In total, 18 missense variants were suspected as likely benign or benign, showing the following characteristics: (1) high allele frequency in the gnomAD database; (2) predicted as benign by more than three prediction tools; (3) individuals with the same variant in our cohort presented with unrelated phenotypes; and (4) lack of co-segregation evidence in published families (Appendix A).

By December 2022, a total of 1013 families were reported to have *PRPH2*-associated retinopathy, and phenotypic variability in affected family members was observed in 40 families. Macular degeneration of various forms was the most commonly reported phenotype in the literature, accounting for 45.2% (458/1013) of families; RP ranked second (27.9%, 283/1013); CACD, CRD, and other rare retinopathies accounted for 12.0% (122/1013), 9.8% (99/1013), and 5.0% (51/1013), respectively (Figure 3B). Overall, 105 published families were of Asian descent, including three families with variable phenotypes, in which RP was the most common, accounting for 55.2% (58/105). Significant statistical differences were observed in the phenotypic spectrum (proportion of RP and MD) between non-Asian families and Asian families included in the literature (*p* = 1.6 × 10^−10^); this was also the case for published non-Asian families and families included in our current study (*p* = 0.02) (Figure 3B)**.**

### 2.5. Genotype-Phenotype Correlation of Variants in PRPH2

The eight pathogenic or likely pathogenic missense variants identified in our cohort were divided into two groups according to the prediction results of in silico prediction tools, allele frequency in the gnomAD database, and whether they had been reported as pathogenic variants in a previous study. Class 1 included four missense variants that were predicted as damaging by at least six of seven in silico tools, were absent in the gnomAD database, and had been reported as causative for *PRPH2*-associated retinal diseases. Three of the four missense variants in Class 1 caused RP, and one (c.658C>T/p.R220W) led to MD in conjunction with a variant in *ROM1*. The remaining four missense variants were classified as Class 2, including three novel missense variants that were predicted to be damaging by six tools but were absent or had an allele number in the gnomAD database and one reported variant that was predicted to be damaging by five tools. Of these four Class 2 missense variants, three were found to cause MD, and one (c.633C>G/p.F211L) was found to cause RP (Figure 4A).

The 149 reported missense variants were evaluated using the assessment method for in-house variants, of which 69/149 (46.3%) were grouped into Class 1 and 36/149 (24.2%) were considered to belong to Class 2. In total, 254 families were identified with the missense variants of Class 1, of which almost half (49.2%, 125/254) were diagnosed with RP and 50.8% (129/254) were reported with MD. The 36 missense variants of Class 2 were identified in 102 families, of which only 30.4% (31/102) had RP and 69.9% (71/102) had MD. Significant statistical differences in the proportions of RP and MD were observed between the two groups (*p* = 0.001) (Figure 4B). In total, 20 of the remaining 44 missense variants were predicted as benign by more than two prediction tools but were considered pathogenic or likely pathogenic variants according to the ACMG/AMP guidelines. Adding 59 families with the above 20 variants, which were predicted as benign by more than two prediction tools, into the original 102 families with Class 2 missense variants brought the family numbers of Class 2 up to 161, and the statistical differences of the phenotypic spectrum compared to that of Class 1 missense variants were still identified: 35.4% (57/161) of families had RP, and 64.6% (104/161) had MD (*p* = 0.006). Of the 105 missense variants of Classes 1 and 2, 90.5% (95/105) were aggregated in the important ID2 domain of proteins, and 54.5% (24/44) of the remaining were located in this domain with significantly statistical differences (*p* = 6.1 × 10^−7^) (Figure 4).

### 2.6. Clinical Characteristics of Published Families with PRPH2 Variants

In the literature, the age of disease onset was available for 606 patients, where 106 patients with RP had symptoms earlier than the 339 patients with MD, showing statistical differences (*p* < 0.0001). Of 161 patients with other retinal phenotypes, almost half of them (44.7%, 72/161) had symptoms after 40 years old. Of the 91 patients with an age of onset of no more than 18, 44/91 (48.4%) were diagnosed with autosomal recessive LCA or RP, while among the 221 patients with an age of onset of more than 18 but no more than 40 and the 294 patients with an age of onset of more than 40, MD was identified in 56.6% (125/221) and 63.9% (188/294), respectively, showing statistical differences (*p* < 0.0001). BCVA data were available for a total of 1070 patients, and the majority of the patients had tests administered after the age of 40. Of the 153 patients with an examination age of no more than 40, 16/153 (10.5%) had a BCVA of no more than 0.3, while 127/573 (22.2%) patients with an examination age of more than 40 had a BCVA of no more than 0.03 with significant statistical differences (*p* = 0.001) (Appendix A). Based on the literature, the clinical characteristics of RP and MD caused by variants in *PRPH2* are summarized in Table 2.

## 3. Discussion

### 3.1. Summary Findings in the Current Study

The current study comprehensively summarized the genotypic and phenotypic spectrum of *PRPH2*-associated diseases in different populations and expanded knowledge concerning genotype-phenotype correlation of *PRPH2*-related conditions based on a large Chinese cohort with inherited retinal diseases and a thorough review of published data. In our cohort, a total of ten variants of *PRPH2* (two truncation and eight missense) were considered pathogenic or likely pathogenic based on ACMG/AMP guidelines. Half of the ten variants caused RP in nine families, and the other half were attributed to MD. The variant c.914del/p.G305Afs*19 was most common in our cohort, different from the most frequently reported variant c.514C>T/p.R172G in previously published data. Furthermore, in previous reports about the phenotypic profile of *PRPH2*, *PRPH2* was found to play a dominant role in MD, but it has not been classified as a common causative gene for RP. Yet RP was more common than MD in our cohort. The nine RP families in our cohort suggested a notable contribution of *PRPH2* to RP, and we found that it might be one of the top six causative genes for autosomal dominant RP. Comparing the proportion of RP and MD, the two primary phenotypes of *PRPH2*, between different classes of missense variants, it was found that patients with missense variants predicted to be damaging by all prediction tools and absent in the gnomAD database were more likely to develop RP. This finding was further confirmed in the reported families, where the proportion of RP was significantly higher in families with missense variants predicted to be damaging by more tools and with no more than one allele number in the gnomAD database.

### 3.2. Exploration of the Genotype-Phenotype Correlation

*RDS* has been extensively studied, as has its function in determining the architecture of photoreceptor outer segments [16,17]. Mice carrying the pathogenic variant of *PRPH2* were reported to have disorganized outer segments, and their photoreceptors were found to be progressively undergoing apoptosis [18,19,20,21]. The high genetic and phenotypic heterogeneity of *PRPH2*-associated retinopathy undoubtedly increases the difficulty of assessing individual *PRPH2* variants. In the inheritance modes, although the autosomal dominant trait was predominant, autosomal recessive and digenic forms with a variant of the homologous *ROM1* gene were also reported, though relatively rare. Pathogenic variants of *PRPH2* were found to be causative of a broad spectrum of both rod- and cone-dominant forms of retinal degeneration, with significant inter- and intrafamilial phenotypic variability, ranging from RP to variable forms of MD; this is consistent with reports on the distinct roles and methods of rod and cone maintenance found in both in vivo and in vitro experiments [22]. In studies conducted on models carrying causative variants of *PRPH2*, it was indicated that loss-of-function variants, i.e., p.C214S and p.P216L, contributed to the rod-dominant retinal disease due to haploinsufficiency, while gain-of-function variants, i.e., p.C150S and p.R172W, caused cone-dominant phenotypes [23,24,25,26,27]. Even so, the genotype–phenotype correlation of *PRPH2*-associated retinopathy remains unclear, and the diagnosis, prognosis, and genetic counseling of variants in *PRPH2* in the clinic are still difficult, especially for those ophthalmologists without extensive knowledge of this gene. In our cohort, 75% of patients carrying *PRPH2* missense variants predicted to be damaging by at least six of the seven computational tools and absent in the general population database were diagnosed with RP. Among patients carrying missense variants predicted to be damaging by five tools and absent in the gnomAD or predicted to be damaging by six computational tools and with an allele count in the gnomAD, 75% were found to have MD. Based on these findings, it was suspected that patients carrying missense variants in *PRPH2* with a higher degree of pathogenicity based on the in silico tools and a lower allele number in the general population database are more likely to present with RP. This hypothesis was verified in previous studies that showed that RP is more common in patients carrying missense variants that were predicted to be damaging by more prediction tools but not available in the general population databases. The genotype–phenotype correlations of *PRPH2*-associated diseases summarized in the current study provide a valuable assessment of *PRPH2* variants to help in the preliminary determination of the pathogenicity of *PRPH2* variants detected in the clinic without functional experiments.

### 3.3. Phenotypic Profile of PRPH2 in Different Populations

Although many studies have been conducted on *PRPH2*-related diseases, most of them focused on Caucasian individuals, and only limited studies have been reported on *PRPH2*-associated diseases in large Chinese populations. It is possible that the frequency of different phenotypes relates to one ethnicity and may be different in another population, as the finding found by us that ethnic background is one of the main influencing factors of *PRPH2*-related phenotypes. In previous studies performed in the Caucasian population, it was found that variants of *PRPH2* contributed less to RP but correlated with autosomal-dominant MD [28,29,30,31,32], which was the same as the phenotypic spectrum of *PRPH2* in the literature review. The low frequency of *PRPH2* variants in Chinese families with autosomal dominant RP has also been reported [33]. In our cohort, pathogenic or likely pathogenic variants of *PRPH2* were found in nine RP families, including eight autosomal-dominant RP families and one recessive trait, revealing a not insignificant contribution among our 1519 RP families. Although the exact number of families with an autosomal-dominant pattern in these 1519 in-house RP families is uncertain, it is considered that *PRPH2* is an important causative gene for autosomal dominant RP. The age of onset for RP patients was notably earlier than MD patients, which might be explained by the larger number of RP patients in our clinic. In addition, although the digenic form of *PRPH2* was originally reported to cause RP, both families in the digenic form found in the current study exhibited MD.

In our cohort, two truncation, five potential pathogenic missense variants, and one in-frame variant were thought to be variants of unknown significance and identified in individuals showing phenotypes unrelated to *PRPH2*. One *PRPH2* missense variant, c.658C>T/p.R220W, identified in an MD patient in the digenic form was also detected in a male over 30 years old who was diagnosed with early-onset high myopia, but no macular dystrophy was observed during fundoscopy and OCT examination, which means it may be caused by other high-myopia-associated causative genes. As the age of onset of MD caused by *PRPH2* variants is usually later than 40 years, follow-up of these patients, who likely carry the pathogenic *PRPH2* variant but lack the relevant phenotype at the time, should be performed. The in-house individuals who had the missense variants predicted as benign by more than two tools and had no less than one allele number in the gnomAD database presented with unrelated phenotypes as well, and variants in *PRPH2* were not co-segregated among some of the available family members. Some reported missense variants were also predicted to be benign by a number of tools, had a high allele frequency in the gnomAD, and lacked co-segregation; furthermore, detailed clinical data in the literature were also considered uncertain. A co-segregation analysis as well as long-term follow-up are recommended for all patients detected with variants in *PRPH2*, regardless of their predicted outcome by prediction tools.

### 3.4. Conlusions

In conclusion, this study systematically researched *PRPH2*-associated retinopathy involving variant features, the clinical spectrum, and genotype–phenotype correlations. The phenotypic profile of *PRPH2* variants is found to be different for different ethnic backgrounds. For example, *PRPH2* was one of the common contributors to autosomal-dominant RP in the Chinese cohort; this information is valuable for the interpretation of *PRPH2* variants in populations with different ethnic backgrounds because it shows an earlier age of onset of RP. Although the exact genotype–phenotype correlation of *PRPH2*-associated retinal diseases is unclear, the exploration of genotype–phenotype correlations in this study provides further information. In the absence of basic experiments, our findings could be of help in the initial clinical diagnosis and interpretation of *PRPH2* variants.

## 4. Materials and Methods

### 4.1. Subject Recruitment and Data Collection

This retrospective study was approved by the Institutional Review Board of Zhongshan Ophthalmic Center. Individuals with different inherited ocular conditions as well as available family members were recruited from our Pediatric and Genetic Clinic, Zhongshan Ophthalmic Center, Guangzhou, China. The informed consent forms adhered to the tenets of the Declaration of Helsinki were signed by probands or their guardians. Clinical data and peripheral venous blood samples from participants and family members were collected. Genomic DNA was extracted from the leukocytes within peripheral venous blood samples as previously described [34].

### 4.2. In-House Variant Evaluation and Phenotype Characterization

Targeted exome sequencing or whole exome sequencing was performed on the genomic DNA of the 10,530 recruited probands in our clinic, including 1519 families with RP and 9011 unrelated probands with other variable eye conditions, using the same methods as those described in our previous studies [35,36]. Variants of *PRPH2* were collected from an in-house exome sequencing dataset and those with a low coverage depth, synonymous variants, with minor allele frequencies of no less than 1% in the gnomAD general population dataset, located in the untranslated region, and the non-canonical splicing sites were excluded. Based on the multi-step bioinformatics analyses described in our previous study [35,37], the missense variants of *PRPH2* were predicted using seven in silico tools, including Rare Exome Variant Ensemble Learner (REVEL; https://sites.google.com/site/revelgenomics/about, accessed on 1 June 2022), Combined Annotation Dependent Depletion (CADD; https://cadd.gs.washington.edu, accessed on 1 June 2022), Sorting Intolerant Form Tolerant (SIFT; https://sift.jcvi.org/, accessed on 1 June 2022), Likelihood Ratio Test (LRT; http://genetics.wustl.edu/jflab/lrt_query.html, accessed on 1 June 2022), Mutation Taster (http://www.mutationtaster.org, accessed on 1 June 2022), Functional Analysis Through Hidden Markov Models (FATHMM; http://fathmm.biocompute.org.uk/, accessed on 1 June 2022), and Protein Variation Effect Analyzer (PROVEAN; https://provean.jcvi.org/, accessed on 1 June 2022). The ACMG/AMP guidelines were used to assess the candidate variants [38]. Individuals with pathogenic variants, likely pathogenic variants, and variants of uncertain significance (VUS) were confirmed by performing Sanger sequencing [39]. A co-segregation analysis for available family members was performed.

The available clinical records and characteristics of patients with pathogenic or likely pathogenic variants of *PRPH2* were thoroughly reviewed, including gender, age of onset, age of examination, first symptoms, and family history. The ophthalmic examinations of these patients were summarized in detail, including BCVA, refractive error, fundus photography, OCT, visual field, and full field ERG according to the international standards of the International Society for Clinical Electrophysiology of Vision (ISCEV).

### 4.3. Literature Review of PRPH2 Variants and Associated Ocular Phenotypes

The term “*PRPH2*” was used as the keyword when performing a literature search with PubMed (https://www.ncbi.nlm.nih.gov/pubmed/, accessed on 1 June 2022), up to December 2022. Variants in *PRPH2* in the HGMD (https://www.hgmd.cf.ac.uk/ac/validate.php, accessed on 6 March 2022) were reviewed until December 2022. All the published literature on *PRPH2* in English and recorded *PRPH2* variants in HGMD were enrolled, and duplicates were excluded. Allele number and frequency, type of pathogenic variants, inheritance mode, clinical diagnosis, and available associated ocular phenotypes of all reported variants were collected and summarized. All reported variants were thoroughly evaluated using the same methods as our in-house variants, which were predicted using the same seven in silico tools and compared with the general population database. Finally, all reported variants were assessed according to the ACMG/AMP guidelines.

### 4.4. Statistical Analysis

SPSS Statistics (Version 25.0; IBM Corp., Armonk, NY, USA) was used for the statistical analysis. The phenotypic spectrum, especially the proportion of RP and MD, was compared (1) among all published non-Asian families in the literature, published Asian families, and in-house families; (2) among reported families caused by variants that were predicted with different degrees of pathogenicity; and (3) among reported individuals with different ages of onset, using the *chi*-square test. The distribution of variants that were predicted with different classes of pathogenicity (outside and inside the ID2 domain) and BCVA among reported patients with different ages of examination were also compared by performing the *chi*-square test. The age of onset between reported patients with RP and MD, respectively, was compared by performing the *Mann–Whitney U* test. A *P*-value of less than 0.05 was used to determine statistically significant differences.

## Figures and Tables

**Figure 1 ijms-24-06728-f001:**
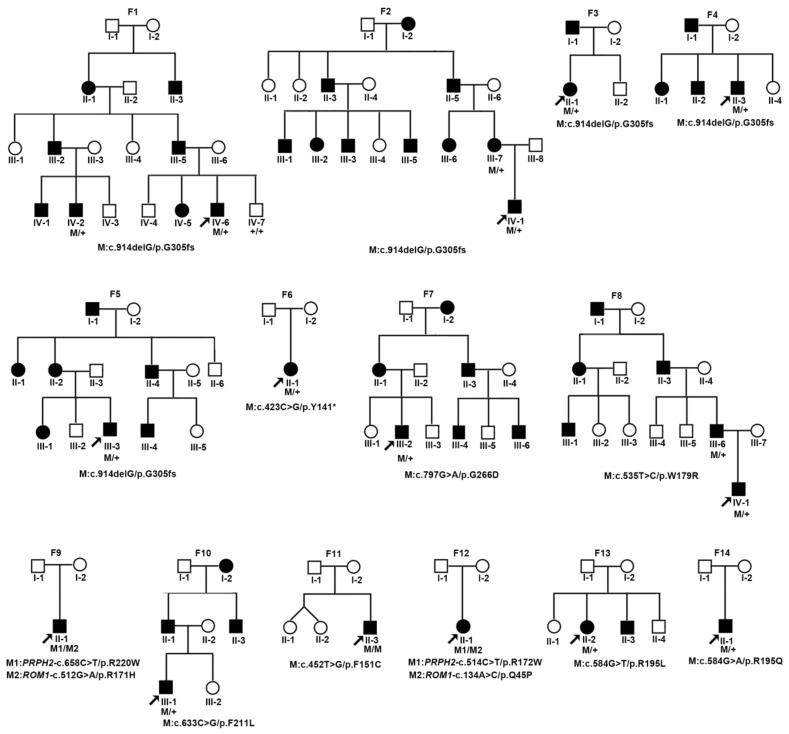
The genealogy of 14 families with pathogenic or likely pathogenic variants of *PRPH2*. Squares represent male individuals, and circles indicate female individuals. Affected patients are indicated by black shading, and the proband of each family is indicated by an arrow. Family numbers are listed on the top of genealogies, while variants are listed under the genealogies.

**Figure 2 ijms-24-06728-f002:**
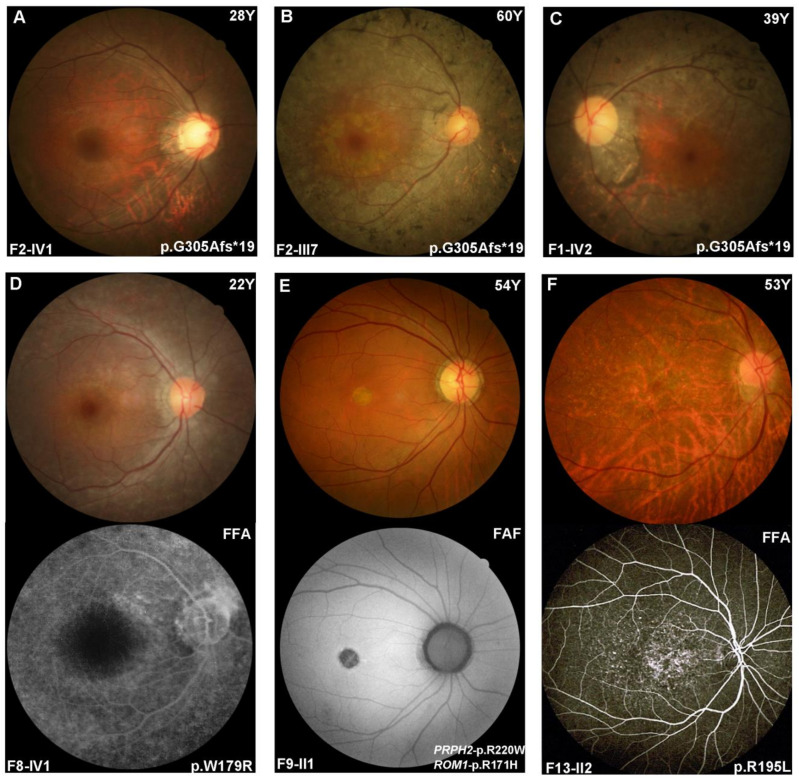
Fundus performance of in-house patients with *PRPH2*-associated retinopathy. (**A**–**D**) Pale optic discs, narrowing of retinal vessels, widespread tapetoretinal degeneration, macular atrophy in different degrees, and bone-spicule pigment deposits in the mid-peripheral area were observed among patients diagnosed with RP. Mottled hyper-fluorescein spots were observed in the whole posterior pole except for the macula area. (**E**) A small round atrophic lesion in the macula that presented as a hypoautofluorescent lesion in the autofluorescence image was observed in a patient with adult vitelliform macular dystrophy. (**F**) A patient presented with granular pattern retinal pigment epithelium mottling diffusely throughout the macular area that showed widespread mottled hyper-fluorescein in the late stage of fluorescein fundus angiography.

**Figure 3 ijms-24-06728-f003:**
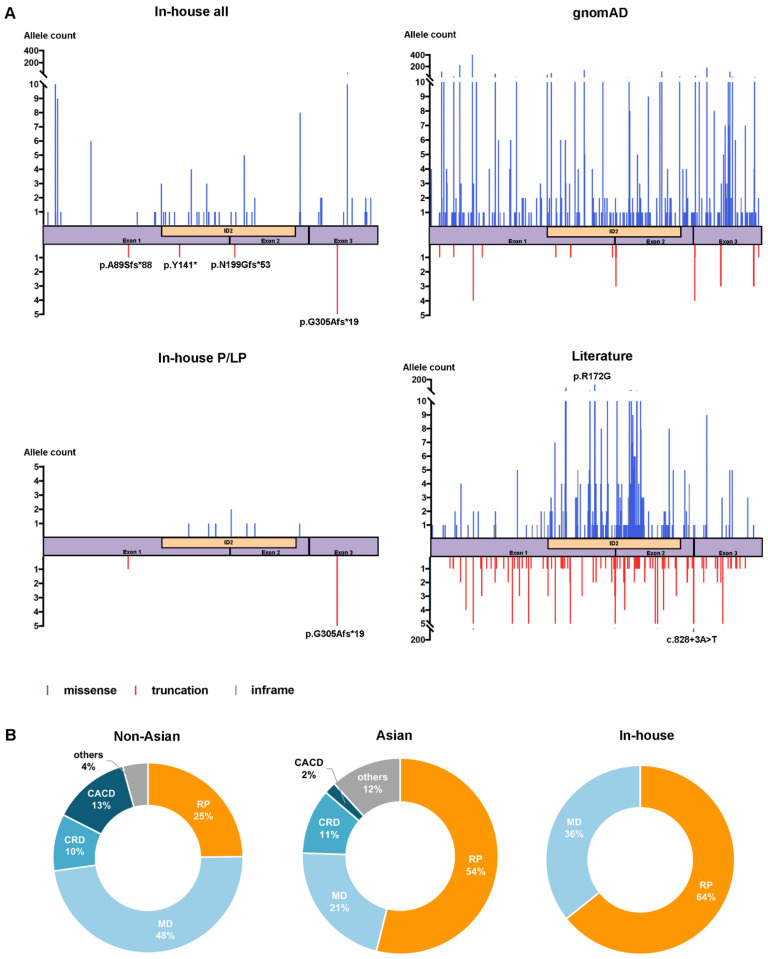
The genotypic and phenotypic spectrum of variants in *PRPH2* in our cohort and reported in the literature. (**A**) Distribution and frequency of all detected *PRPH2* variants in our cohort, the gnomAD database, pathogenic or likely pathogenic *PRPH2* variants in our cohort, and reported *PRPH2* variants in the literature. The blue, red, and gray lines represent missense, truncation, and in-frame variants, respectively. ID2 indicates the intradiscal D2 loop of the peripherin protein. (**B**) The three pie charts show the phenotypic spectrum of families with *PRPH2* variants identified in our cohort, Asian families, and all reported non-Asian families with *PRPH2*-associated retinopathy in the literature, respectively.

**Figure 4 ijms-24-06728-f004:**
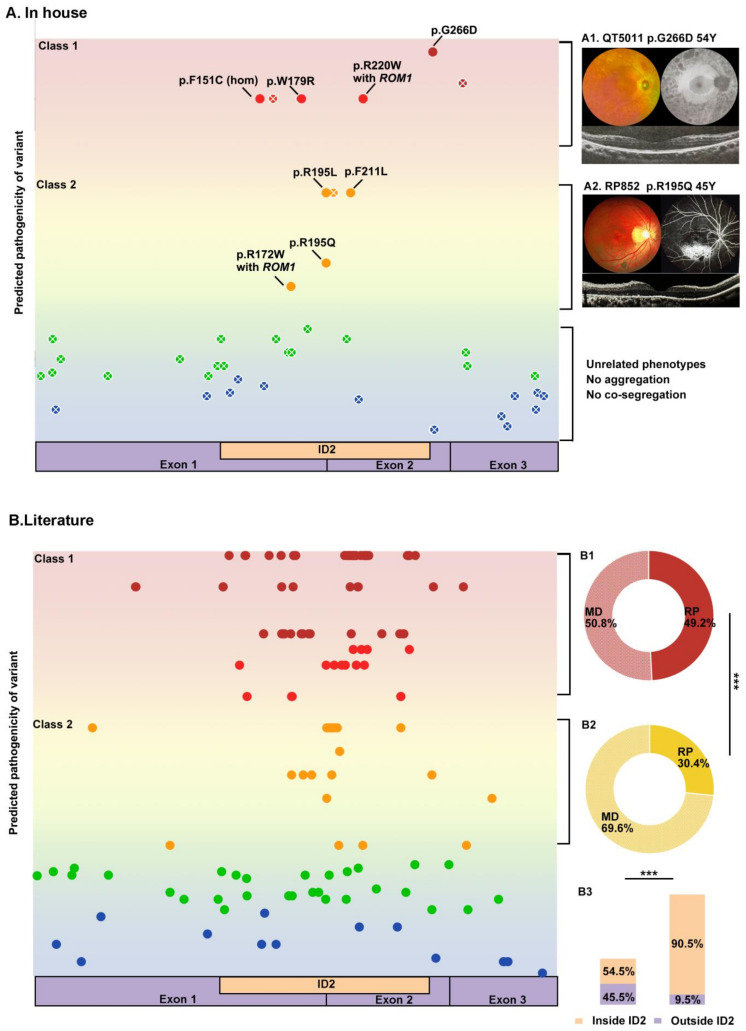
The exploration of the correlation between genotype and phenotype of *PRPH2*-associated diseases in this study. (**A**) In our cohort, among the four patients carrying missense variants that were predicted by at least six out of the seven in silico tools and absent in the gnomAD database (Class 1), three were diagnosed with retinitis pigmentosa (**A1**). Among the remaining four patients carrying missense variants predicted as damaging by five tools and absent in the gnomAD database or predicted as damaging by six tools and containing one allele number in the gnomAD database, three were identified with macular degeneration (**A2**). Patients carrying missense variants predicted to be benign by more than three tools and with more than one allele number in the gnomAD database presented phenotypes unrelated to *PRPH2* that were not co-segregated among family members and were not aggregated among individuals with inherited retinal dystrophy compared to controls. (**B**) In total, 49.1% of the 254 families carrying Class 1 missense variants were diagnosed with retinitis pigmentosa (**B1**), while 69.6% of the 102 families with Class 2 missense variants were diagnosed with macular degeneration (**B2**), which was statistically significant (*p* = 0.001). Among all of the 149 reported variants, 90.5% of the Class 1 and Class 2 missense variants were located in the ID2 loop domain of proteins, and only 54.5% of the remaining variants were found to aggregate in the domain (**B3**). *** represented *p* ≤ 0.001.

**Table 1 ijms-24-06728-t001:** Genotypic details of seventeen potential pathogenic variants in *PRPH2* identified in our cohort.

No	Position in chr6	Change	Effect	FN	ACMG Evidence	ACMG Rank	①	②	③	④	⑤	⑥	⑦	gnomAD AF	HGMD
**Truncation variants in *PRPH2***															
1	42666162	c.914del	p.G305Afs*19	5	PVS1, PS4, PM2, PP1, PP4	P	/	/	/	/	/	/	/	/	DM
2	42672324	c.595_607del	p.N199Gfs*53	1	PVS1, PM2, BP5	VUS	/	/	/	/	/	/	/	/	/
3	42689650	c.423C>G	p.Y141*	1	PVS1, PS4, PM2, PP4	P	/	/	/	/	/	/	/	/	/
4	42689808	c.264dup	p.A89Sfs*88	1	PVS1, PM2, BP5	VUS	/	/	/	/	/	/	/	/	/
**Missense variants in *PRPH2* (Class 1)**															
5	42666217	c.857T>C	p.L286P	1	PM2, PP3	VUS	0.904	25.9	D	D	D	D	D	/	DM?
6	42672134	c.797G>A	p.G266D	1	PS1, PS4, PM2, PP3, PP4	P	0.92	26.3	D	D	D	D	D	/	DM
7	42672273	c.658C>T	p.R220W	2	PS4, PM2, PP3, PP4	LP	0.824	25	D	D	D	D	D	/	DM
8	42689538	c.535T>C	p.W179R	1	PS1, PS4, PM2, PP1, PP3, PP4	LP	0.898	29.3	D	D	D	T	D	/	DM
9	42689540	c.533A>G	p.Q178R	1	PM2, PP3	VUS	0.875	26.6	D	D	D	T	D	/	DM
10	42689595	c.478C>A	p.Q160K	1	PM2, PP2, PP3, BS4	VUS	0.89	27.5	D	D	D	D	D	/	/
11	42689621	c.452T>G	p.F151C	1	PS4, PP2, PP3, PP4	LP	0.741	28.7	D	D	D	D	D	/	/
**Missense variants in *PRPH2* (Class 2)**															
12	42672298	c.633C>G	p.F211L	1	PS1, PS4, PM2, PP3, PP4	P	0.77	23.9	D	D	D	D	D	/	/
13	42672332	c.599T>G	p.V200G	1	PS1, PM2, PP3, BS2	VUS	0.483	32	D	D	D	T	D	/	DM
14	42672347	c.584G>A	p.R195Q	1	PS1, PS4, PM2, PM5, PP3, PP4	P	0.761	32	D	D	D	T	N	/	DM
15	42672347	c.584G>T	p.R195L	1	PS1, PS4, PM5, PP3, PP4	P	0.618	32	D	D	D	T	D	3.98 × 10^−6^	/
16	42689559	c.514C>T	p.R172W	1	PS1, PS3, PS4, PM1, PM2, PM5, PP3, PP4	P	0.67	26	D	D	D	T	D	/	/
**In-frame variant in *PRPH2***															
17	42689568	c.499_504dup	p.G167_N168dup	1	PM2, BP3	VUS	/	/	/	/	/	/	/	/	/

Notes: ① REVEL; ② CADD; ③ SIFT; ④ LRT; ⑤ MutationTaster; ⑥ FATHMM; ⑦ PROVEAN; AF = allele frequency; FN = family numbers; P = pathogenic; LP = likely pathogenic; VUS = variants uncertain significance; D = damaging; T = tolerant; N = neural; DM = damaging mutations.

**Table 2 ijms-24-06728-t002:** Clinical characteristics of patients with *PRPH2*-associated retinopathy.

Characteristic	*PRPH2*-Associated RP (n = 429)	*PRPH2*-Associated MD (n = 720)
**Gender**		
Male	160	228
Female	121	319
**Age at onset (years)**		
Mean ± SD	30.1 ± 17.2 (n = 100)	41.5 ± 14.1 (n = 307)
Range	6.0–17.0	3.0–75.0
**Age at exam (years)**		
Mean ± SD	49.3 ± 17.7 (n = 155)	51.2 ± 14.6 (n = 472)
Range	8.0–86.0	6.0–90.0
**BCVA**		
BCVA < 0.05	9/88	17/408
0.05 ≤ BCVA < 0.3	15/88	46/408
BCVA ≥ 0.3	64/88	345/408
**Fundus change**	Typical RP performance	Variable MD performance (PD, STGD…)
**ERG**	Severe rod involvement is predominant	Normal or mild cone involvement
**OCT**	Macula is relatively spared Periphery thinning of the outer retinal layer	Hyperreflective deposit above the RPE in the foveal/perifoveal regions
**Type of variants**		
Truncation	69	296
Missense	360	423

Notes: SD = standard deviation; BCVA = best-corrected visual acuity; RP = retinitis pigmentosa; MD = macular degeneration; PD = pattern dystrophy; STGD = Stargardt disease; ERG = electroretinogram; OCT = optical coherence tomography.

## Data Availability

The data presented in this study are available on request from the corresponding author. The data are not publicly available due to ethical privacy.

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
