# Peer review of "New Insight into the Genotype-Phenotype Correlation of PRPH2-Related Diseases Based on a Large Chinese Cohort and Literature Review"

_ijms, 2023, doi:10.3390/ijms24076728_

Round 1

Reviewer 1 Report

Provided in the attached Word document

Reviewer 2 Report

Title:  New insight into the genotype-phenotype correlation of PRPH2 based on a large Chinese cohort and literature review.

Authors: Wang, Y., Wang, J., Jiang, Y., Zhu, D., Ouyang, J., Yi, Z., Li, S., Jia, X., Xiao, X., Sun, W., Wang, P., and Zhang, Q.

Summary: This paper describes the analysis of data that identified PRPH2 variants. PRPH2 variants are a common cause of inherited retinal dystrophies. This paper demonstrates the importance of evaluating PRPH2 variants in distinct population, as the data suggests that the frequency is higher in the Chinese cohort and that this can be used as a predictive tool to identify inherited retinal dystrophies.

Revisions needed:

Page 1, Line 11  - Insert a after the word were.

Page 1, Line 11 – change the word highly to high.

Page 1, Line 12 – change the word heterogenous to heterogeneity.

Page 1, Line 13 – insert the word the before the word assistance.

Page 1, Line 16 – change the word sequence to sequencing.

Page 1, Line 22 – change absence is gnomAD to absent in the gnomAD.

Page 1, Line 23 – insert the word the before gnomAD.

Page 1, Line 41 – change the word severer to more severe.

Page 1, Line 43 – change accurate diagnosis in clinical to accurate clinical diagnosis (delete in clinical).

Page 1, Line 44 – insert the word a before large.

Page 2, Line 47 – change the word cohort to cohorts.

Page 2, Line 48 – change the word cohort to cohorts (after Asian and after Chinese).

Page 2, Line 49 – change the word have to has.

Page 2, Line 50 – change Spanish to Spain.

Page 2, Line 54 – change the word comparative to compared.

Page 2, Line 61 – insert the word the before Chinese.

Page 2, Line 62 – change recently to recent.

Page 2, Line 63 – delete the word detailly.

Page 2, Line 66 – delete the word was after Asian and insert the word subjects (Asian subjects revealed).

Page 2, Line 66 – change mainly to main.

Page 2, Line 66 – insert the word this before Asian.

Page 2, Line 67 – change Caucasian to Caucasians.

Page 2, Line 69 – change precited to predicted.

Page 2, Line 70 – change phenotype to phenotypes.

Page 2, Line 71 – change suggestions to suggestion.

Page 3, Line 90 – change assessed to assessing.

Page 4, Line 101 – change causing to caused.

Page 4, Line 111 – insert the word an before allele.

Page 5, Line 116 - change puts to are listed.

Page 5, Line 120 – change in after families to with, change in to with the before autosomal recessive.

Page 5, Line 121 – change vision decreased to decreased vision; insert the before third.

Page 5, Line 123 – change BCVA recording in whom to BCVA recordings in which.

Page 6, Line 133 – change narrow to narrowing.

Page 6, Line 135 – insert the word in before the.

Page 6, Line 136 – insert the word the before macula area; insert the word the before macula that.  

Page 6, Line 137 – change image to images.

Page 7, Line 151 – change thinner to thinning.

Page 7, Line 155 – change population to populations

Page 7, Line 156 – insert the before literature.

Page 7, Line 159 – change 5-untranslated to 5’-untranslated.

Page 7, Line 160 – change frequency to frequent.

Page 7, Line 162 – change frequency to frequent.

Page 7, Line 166 – change assistant to assistance.

Page 7, Line 167 – insert the before gnomAD.

Page 7, Line 178 – delete families before the word published.

Page 7, Line 181 – change the word that to for.

Page 8, Line 185 – delete that.

Page 8, Line 186 – insert the before literature.

Page 9, Line 191 – change assocaited to associated.

Page 9, Line 196 – insert a before previous.

Page 9, Line 197 – insert were after that.

Page 9, Line 202 – change to variants that were predicted to be damaging.

Page 9, Line 211 – remove ) after MD.

Page 9, Line 216 – remove ( before 33. 3%.

Page 9, Line 225 – change of 56.6% to was 56.6%.

Page 9, Line 227 – change to mostly to most, insert of the after most, delete were after patients. (most of the patients presented…)

Page 9, Line 229 – insert the word who before had.

Page 9, Line 230 – change significantly to significant, change statistic to statistical.

Page 11, Line 240 – change co-segregation to co-segregated.

Page 11, Line 242 – deleted with after carried.

Page 11, Line 244 – delete the after among, insert of after all (among all of the….).

Page 11, Line 251 – change thoroughly to thorough.

Page 11, Line 261 – delete carried after patients.

Page 11, Line 264 – delete carrying after families.

Page 11, Line 271 – insert the word  undergoing before apoptosis.

Page 11, Line 283 – change in clinical to in the clinic.

Page 11, Line 284 – change didn’t well known to didn’t have much knowledge.

Page 12, Line 286 – delete that before 75%.

Page 12, Line 295 – insert the before general.

Page 12, Line 298 – insert the before clinic.

Page 12, Line 299 – change have to has.

Page 12, Line 305 – insert the before literature.

Page 12, Line 319 – change though to thought.

Page 12, Line 321 – change were to was after that, change were to was after form.

Page 12, Line 323 – change were to was before likely, change gene to genes.

Page 12, Line 324 – change were to was after dystrophy.

Page 12, Line 326 – insert the word to before pay, insert to this after attention.

Page 12, Line 332 – insert the before literature.

Page 13, Line 340 – change An before informed to The.

Page 13, Line 341 – after Helsinki change was to and were.

Page 13, Line 366 – change that were to including (before gender).

Page 14, Line 397 – after study, delete was a, delete research of  and insert researched (study systematically researched PRPH2….)

Page 14, Line 400 – change background to backgrounds, change contributor to contributors.

Page 17, Line 550 – deleted 1.
